# Switching at the ribosome: riboswitches need rProteins as modulators to regulate translation

Vanessa de Jesus [1], Nusrat S. Qureshi[1], Sven Warhaut[1], Jasleen K. Bains[1], Marina S. Dietz [2], Mike Heilemann [2], Harald Schwalbe [1] & Boris Fürtig [1✉]

Translational riboswitches are cis-acting RNA regulators that modulate the expression of genes during translation initiation. Their mechanism is considered as an RNA-only gene-regulatory system inducing a ligand-dependent shift of the population of functional ON- and OFF-states. The interaction of riboswitches with the translation machinery remained unexplored. For the adenine-sensing riboswitch from *Vibrio vulnificus* we show that ligand binding alone is not sufficient for switching to a translational ON-state but the interaction of the riboswitch with the 30S ribosome is indispensable. Only the synergy of binding of adenine and of 30S ribosome, in particular protein rS1, induces complete opening of the translation initiation region. Our investigation thus unravels the intricate dynamic network involving RNA regulator, ligand inducer and ribosome protein modulator during translation initiation.

[1] Institute of Organic Chemistry and Chemical Biology - Center for Biomolecular Magnetic Resonance (BMRZ), Johann Wolfgang Goethe-University Frankfurt, Max-von-Laue Straße 7, Frankfurt am Main, Germany. [2] Institute of Physical and Theoretical Chemistry, Johann Wolfgang Goethe-University Frankfurt, Max-von-Laue Straße 7, Frankfurt am Main, Germany. ✉email: fuertig@nmr.uni-frankfurt.de

During translation initiation, messenger RNA (mRNA) and the 30S ribosome form a complex that is stabilized by interactions between mRNA, ribosomal RNA (rRNA) and ribosomal proteins (rproteins). The mRNA contains a ribosome binding site (RBS) of approx. 30 nucleotides (nt) around the start codon that includes the evolutionary conserved Shine-Dalgarno (SD) sequence 5'-GGAGGA-3'[1,2]. Translation initiation requires the coupling of the opening of mRNA secondary structure in the translation initiation region and binding of the 30S ribosome[3]. The opening of mRNA secondary structure is transient and often too short for recruiting the 30S ribosome[4]. To circumvent this short opening, the 30S is already bound to an mRNA standby site[4–7] and shifts into place to form a specific complex with the SD-mRNA and the anti-SD-rRNA interaction only if the SD-mRNA site becomes accessible.

Sensing of the accessibility of the SD sequence thus is one mechanism to regulate translation initiation[8]. In fact, translational riboswitches, cis-acting regulation elements that bind to a specific inducer ligand found in bacteria, modulate this accessibility[9,10]. Up to now, it is assumed that binding of an inducer ligand is sufficient for translational riboswitches to interfere with the dynamic shift between non-specific standby binding and specific binding of the 30S[9,11]. The adenine-sensing *add* riboswitch (ASW) from the human pathogenic bacterium *Vibrio vulnificus* regulates translation in an inducer ligand-dependent (characterized by the dissociation constant $K_D$) and temperature-compensated (characterized by the equilibrium constant $K_{pre}$) manner and functions over a broad temperature range[12,13]. In the absence of adenine, ASW populates two secondary structures (apoB and apoA) (Fig. 1, Supplementary Fig. 1). Both, apoB and apoA are functional OFF-states as they sequester the SD in stable base pair interactions that preclude translation initiation. The *add* riboswitch can bind adenine post-transcriptionally and operates under thermodynamic control, as shown by in vitro and in vivo studies[14]. Gene regulation of the *add* riboswitch is independent from the coupling of transcription and translation. It is clear that the kinetics of ligand binding

and structural switching between apoA and apoB has to occur on a timescale faster than translation initiation[12,15]. It was shown that the rate limiting step in regulation is dependent on the RNA refolding rate, which is independent from adenine concentration[12]. Little is known about the interaction of translational riboswitches and the ribosome, since X-ray crystallography or cryo-EM studies are difficult as the regulatory relevant interactions are transient. However, NMR studies capable of detecting such transient interactions are challenging because of the size of the investigated system of more than 800 kDa, which contains the riboswitch mRNA and the 30S ribosome. Here, we succeed in applying solution NMR spectroscopy of various reconstituted adenine-sensing riboswitch-30S ribosome complexes to investigate both the influence of the inducer ligand adenine and the ribosomal modulator protein rS1 and unravel a synergistic regulatory switch to a functional ON-state.

## Results and discussion

We investigated the riboswitch (ASW[14–140]) containing 18 nucleotides within the coding region of the mRNA as a standby site for ribosome binding (Fig. 1a). The inducer ligand adenine binds to ASW with a $K_D$ of 0.6 µM at 25 °C[12]. The helix P4 of ASW[14–140] was only partially opened up to 20% at 25 °C, and complete opening was never reached even at saturating concentrations of the inducer ligand (Fig. 1b, Supplementary Fig. 1).

The footprint of bound 30S ribosome comprises 30 nucleotides that start 15 nt upstream of the start codon (position −15)[16]. The adenine-induced opening of RNA structure in ASW[14–140], however, only affected nucleotides up to position −9, nucleotide A111 (Supplementary Fig. 1). This ligand-induced partial opening of P4 in ASW[14–140] was not sufficient to initiate translation. In other words, the interaction of adenine alone with ASW[14–140] was insufficient to enable the complete accommodation of the mRNA into the ribosome, as the neighboring helix P5 ultimately impeded binding. Therefore, we here determined the additional factors

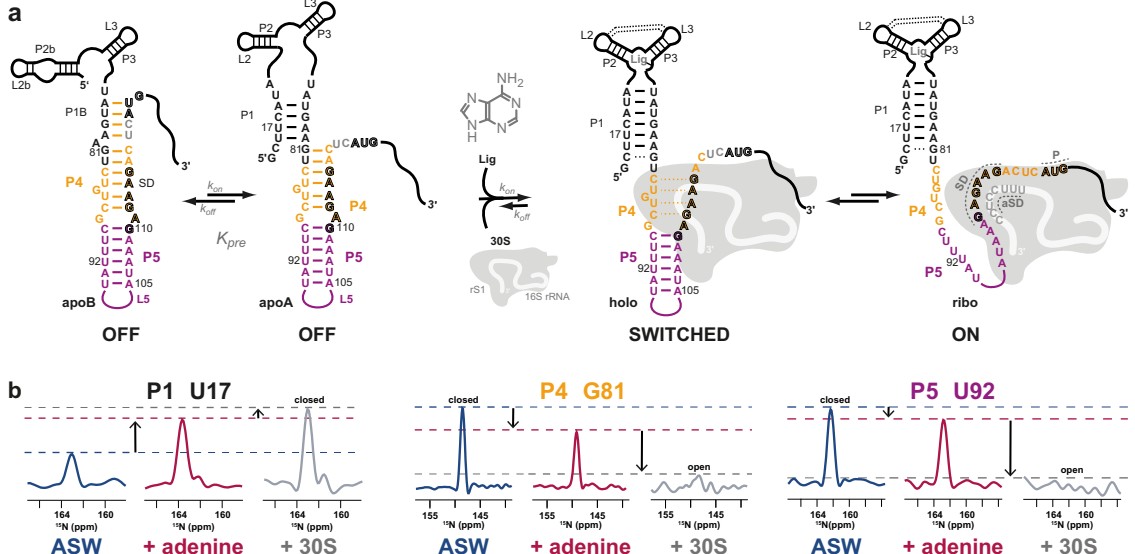

**Fig. 1 Adenine-sensing riboswitch (ASW) regulating translation initiation in the context of adenine and 30S ribosome binding. a** The two functional OFF-conformations apoB and apoA are in a temperature-dependent equilibrium linked through $K_{pre}$. Adenine binding to the aptamer region leads to a compact structure with tertiary interactions such as the kissing loop interaction between L2 and L3. 30S ribosome binding to the standby site 3′-terminal to the AUG start codon of the mRNA results in to melting of the Shine-Dalgarno (SD) sequence in helix P4 of the riboswitch (switched). Further, a stable SD-aSD interaction forms and the mRNA is incorporated into the mRNA tunnel when both P4 and P5 are opened. **b** 1D projections of the 15N dimension of 2D [1H-15N]-BEST-TROSY experiments at 25 °C shown for peaks in helices P1, P4 and P5 for ASW[14–140] without (apo, blue), with 1.4 eq. adenine (holo, magenta) and in complex with the ribosome (30S + holo, gray).

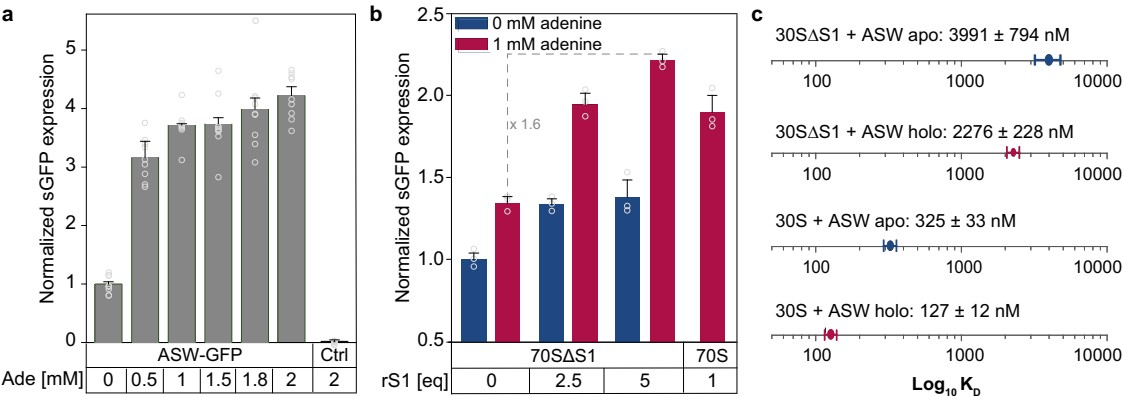

**Fig. 2 Biochemical probing of ASW function in vivo and in vitro. a** Adenine-dependent riboswitch function in vivo. Translation efficiency was monitored by quantifying sGFP expression levels under the control of the riboswitch. For error estimation, three biological independent samples were measured in triplicates (mean ± s.d.). **b** Riboswitch function in a fully reconstituted coupled transcription-translation assay in dependence of adenine and rS1. For error estimation, one biological sample was measured in triplicates (mean ± s.d.). **c** Binding affinities of mRNA-ribosome-complexes determined by microscale thermophoresis in the absence and presence of rS1 and adenine. Source data are provided as a Source data file.

that interact with the riboswitch to enable full accommodation in the ribosomal tunnel.

**rS1 modulates riboswitch activity.** When inserted prior to the shifted green fluorescent protein (sGFP) gene, the riboswitch retained its regulatory function in vivo. Expression of sGFP under the control of the ASW led to an adenine-dependent increase of translated sGFP levels (Fig. 2a). Adenine concentrations in the range between 0.5 mM and 2 mM increased sGFP expression by ~3- to ~4-fold, respectively.

Since, the inducer ligand adenine alone could not account for a structural transition to the functional ON-state, the results of the in vivo experiments suggested that there must be additional factors of the bacterial translation machinery that positively interfered with the riboswitch. Several ribosomal proteins exhibit ATP-independent RNA chaperone activity that accelerates opening of RNA secondary structure[17,18]. Chaperone activity has been reported for rproteins rS12[17,19], and in particular rS1[20]. rS1 is essential for translation initiation in Gram-negative bacteria[21]. Bound to the 30S ribosome[22–25], it recruits mRNAs for translation initiation by binding to AU-rich sequences. We thus investigated the functional role of rS1 in assisting riboswitch-mediated gene regulation. rS1 concentrations cannot be easily manipulated in vivo[26], as knock-out of rS1 is lethal[27]. Thus, we investigated its role in a reconstituted coupled transcription-translation assay (Fig. 2b) to determine rS1's function in the riboswitch-mediated regulation of gene expression. In line with the in vivo experiments, addition of adenine resulted in an increase of sGFP expression in the presence of rS1. However, the regulation efficiency diminished to basal levels in the absence of rS1, decreasing by a factor of 1.6 upon removal of rS1. The switching behavior could be restored upon addition of rS1.

Native composite gel electrophoreses revealed complex formation of ASW[14–140] with 30 S ribosomes (Supplementary Fig. 2). As the cognate ribosome from *V. vulnificus* is not accessible for practical and biosafety reasons, the interaction is studied here with the homologous *E. coli* ribosome. The 3′-end of the 16S rRNA in *E. coli* which harbors the aSD sequence is nearly identical (identity = 98%) to that of *V. vulnificus* (Supplementary Fig. 11) supporting the validity of using *E. coli* ribosomes to study the molecular basis of translation initiation for the *add* riboswitch from *V. vulnificus*. Binding of ASW[14–140] to the 30S ribosome was dependent on the presence of rS1 as shown by microscale thermophoresis (MST) (Fig. 2c, Supplementary Fig. 3). The

binding constants of apo and holo ASW[14–140] to 30SΔS1 ribosomes were $K_D = 4000 ± 800$ nM and $2200 ± 200$ nM, respectively. By stark contrast, binding of ASW[14–140] to 30S ribosomes was substantially increased ($K_D = 320$ nM (apo) and 130 nM (holo)). The affinity increase was nearly four-fold larger in the holo-state hinting at synergistic structural effects of the rS1 protein and the inducer ligand adenine (Supplementary Fig. 4). Notably, the affinity of the riboswitch was further increased to $K_D^{30SΔS1+vvrS1} = 60$ nM (holo) when the rS1 protein from *E. coli* was exchanged to rS1 from *V. vulnificus* (Supplementary Fig. 3b). To show that rS1 acts within the 30S complex, the $K_D$ of *V. vulnificus* rS1 towards 30S was determined to be 170 nM and further NMR experiments reported on a tight binding to the 30S ribosome (Supplementary Fig. 5). The observation that adenine binds more weakly to the ASW than the ribosome hints at a structural and functional synergistic effect of the inducer ligand and the ribosome.

**30S ribosome-bound rS1 opens expression platform in holo state.** To delineate the structural basis for the increased affinity, we characterized conformational changes induced by stoichiometric amounts of rS1 and the 30S ribosome using solution NMR spectroscopy to map the base pairing interactions within ASW[14–140] in eight different experimental setups (Fig. 3a, Supplementary Fig. 6). In the absence of inducer ligand adenine (apo state), the conformations apoB and apoA were found to be in equilibrium. In both conformations, helices P4 and P5 were closed. This structural behavior was generally maintained irrespective of the presence of rS1 alone or in the presence of the ribosomal particle without rS1 (30SΔS1). In a titration with rS1, minor effects towards a destabilization of base pairs in the helices P4 and P5 in the translation initiation region were detected at saturating levels of rS1 (Supplementary Fig. 7). Although, in absence of inducer ligand the expression platform was unresponsive against rS1 alone and 30SΔS1, it was slightly responsive against the full ribosomal particle, however the closed conformational state was still the dominant conformation (Fig. 3b). In the presence of adenine (holo state), the equilibrium was shifted away from apoB towards the holo conformation with partial opening of helix P4 and closed helix P5. By stark contrast to the apo state, the holo state of the riboswitch was much more susceptible to conformational changes upon interaction with rS1 and 30S. At 1:1 stoichiometry, addition of rS1 led to an opening of P4 and P5 by 50%. However, addition of 30S containing rS1 led

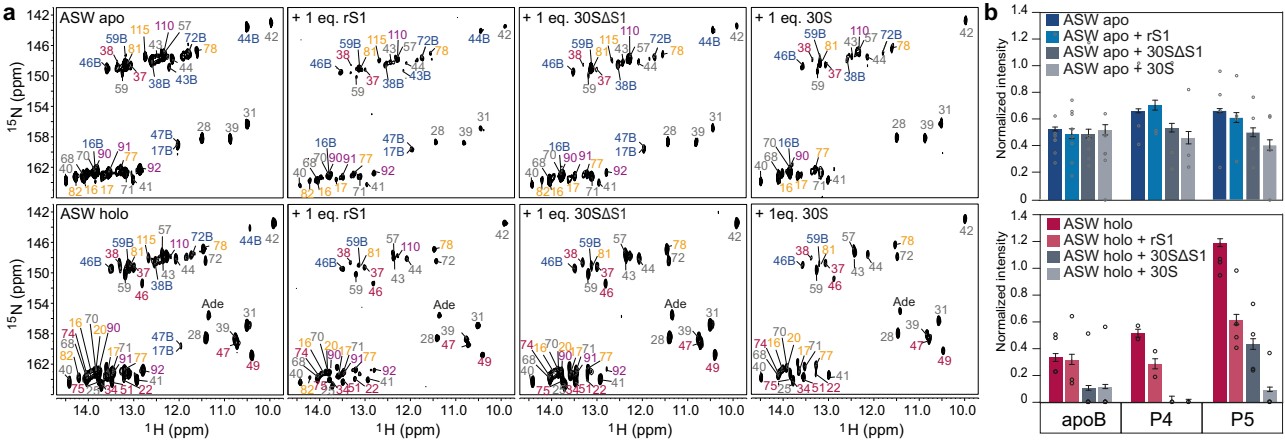

**Fig. 3 NMR spectroscopic investigations of mRNA-ribosome complexes. a** [$^1$H-$^{15}$N]-BEST-TROSY experiments of ASW, ASW-rS1, ASW-30SΔS1 and ASW-30S in absence and presence of 1.4 eq. adenine. Peaks are color coded according to Fig. 1. **b** Normalized intensities of peaks according to the secondary structure elements. Peaks were normalized to helix P3 (see Supplementary Fig. 6, Source data are provided as a Source data file). Error bars represent the standard deviation of the average weighted by the signal-to-noise ratio of the individual peaks.

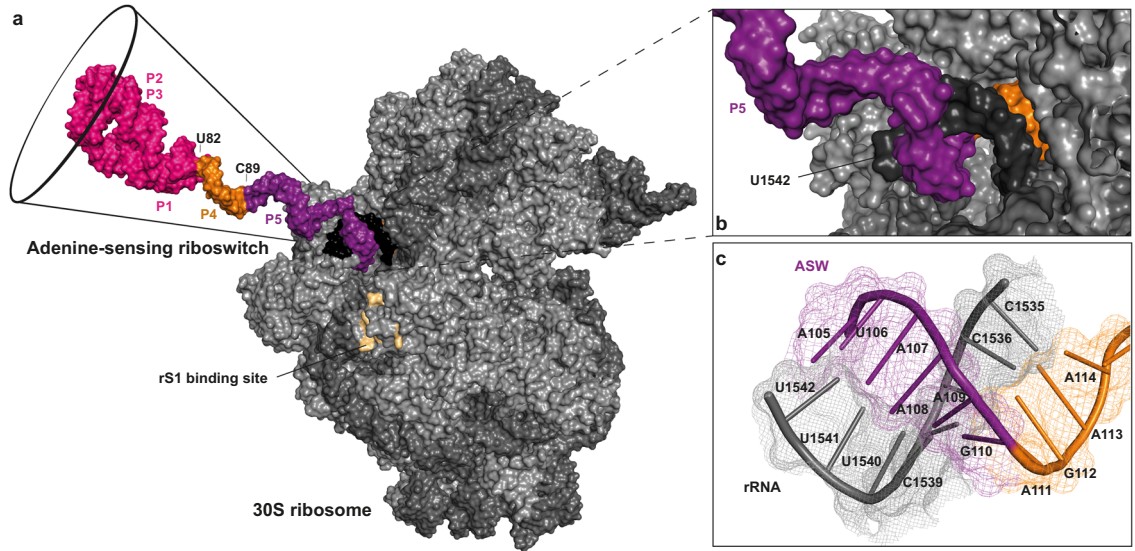

**Fig. 4 Structural model of the adenine-sensing riboswitch (ASW) in complex with 30S ribosome from *E. coli*. a** Overall depiction of the initiating riboswitch with the 3'-end of the 16S rRNA which is shown in dark gray. The structural modules of the ASW are color-coded according to Fig. 1: aptamer (P1, P2, P3) in magenta, 5'-strand of helix P4 in orange, nucleotides of opened stem P5 in purple. The residues of rS2 forming the binding site for rS1 at the 30S ribosome are highlighted in beige[23,30]. **b**, **c** Base pairing interaction of the Shine-Dalgarno sequence of the ASW with the anti-Shine-Dalgarno sequence of the 16S rRNA.

to complete destabilization of the apoB conformation, complete opening of helices P4 and P5 and release of nucleotides 82–115 to adopt single strand conformation.

We built a molecular model of the initiating ribosome in complex with the 5'-end of the ASW mRNA, restrained by the base pairing structure derived from the NMR experiments (Fig. 4). The effect of rS1 on opening of the translation initiation site (P4, P5 and the single stranded 3' nucleotides) was accompanied by a stabilization of base-pairs stemming from P1, P2 and the connecting loops and junctions of ASW[14–140], therefore the ligand bound aptamer was modeled in its compact tertiary structure. Base pairs from position 82 to 115 (P4 and P5) were molten by the 30 S complex and thus the nucleotides were kept in single stranded conformation. Consequently, the RBS including the SD sequence and neighboring secondary structure elements impeding productive mRNA binding becomes accessible for interaction with the 16S rRNA (Fig. 4c). The model showed

that only if helix P5 was single stranded steric clash between r-proteins and mRNA could be avoided at the entrance of the tunnel. At this point, melting of P5 is a unique feature that only occurs through ribosome-bound rS1.

Here, rS1 exerts its RNA chaperoning function[20] directly at the ribosome, fulfilling its role in mRNA delivery by keeping the ribosome binding site on the mRNA in a single-stranded conformation. As it is shown that rS1 promotes partial mRNA unfolding also in other riboswitches[28], facilitating single stranded conformations within the effector domains of translational riboswitches might be a general function of rS1. However, for the ASW, the observed unfolding of this part of the riboswitch is prerequisite for productive incorporation of the mRNA into the 30S ribosome.

In summary, we found that for translational riboswitches binding of the inducer ligand stabilizes the aptamer domain and destabilizes the sequestering effect of the cis-acting

complementary strand precluding SD opening. However, the presence of ligand and riboswitch RNA alone was insufficient to completely release the ribosome binding site. To achieve complete opening, the interaction with the translation machinery and in particular with the rS1 protein was essential. A translational ON-state in the complex between the riboswitch and the 30S ribosome became effectively populated only when both, the inducer ligand and the modulator rS1 were present. Only then, the SD sequence is completely released and could base pair with the aSD sequence of the 16S rRNA leading to correct positioning of the start codon. Thus, our model delineates all required components for translation initiation by riboswitches. We note that our data for the adenine-sensing riboswitch are consistent with the standby model of translation initiation[4,5,29]. Further, it challenges the "RNA-only" model of riboswitch function, as translational regulation only occurs upon concerted yet dynamic interaction of mRNA, 30S ribosomes and associated modulator proteins.

## Methods

**DNA template**. The DNA template of 127 nt adenine-sensing riboswitch (ASW) from *Vibrio vulnificus* is flanked by two restriction sites and two ribozymes at both ends in order to maintain 3′- and 5′-end homogeneity:

GAATTC (EcoRI) – TAATACGACTCACTATAG (T7 Pol) – GGTATGAAGCCTGATGAGTCCGTGAGGACG AAAGACCGTCTTCGGACGGTCTC (Hammerhead ribozyme) – GCTTCATATAATCCTAATGATATGGTTT GGGAGTTTCTACCAAGAGCCTTAAACTCTTGATTATGAAGTCTGTCGCTTTATCCGAAATTTTATAAAGAGAAGACTCATGAATTACTTTGACCTGCCG (127 nucleotide ASW) – GGGTCGGCATGGCATCTCCACCTC CTCGCGGTCCGACCTGGGCTACTTCGGTAGGCTAAGGGAGAAG (HDV ribozyme) – CCCGGG (SmaI).

**Riboswitch preparation**. $^{13}$C–$^{15}$N-labeled 127-nucleotide *add* ASW was synthesized by in vitro transcription with T7 RNA polymerase from linearized plasmid DNA[31]. The $^{13}$C–$^{15}$N-labeled (rATP, rCTP, rGTP, rUTP) nucleotides for transcription were purchased from Silantes (Munich, Germany). The construct was purified by preparative polyacrylamide gel electrophoresis according to standard protocols[32]. The construct was folded in water for 5 min at 95 °C and immediately diluted 10-fold with ice-cold water. The RNA was buffer-exchanged into NMR buffer (25 mM potassium phosphate, 150 mM KCl, 5 mM MgCl$_2$, pH 7.2). For protein-RNA studies, 5 mM DTT was added.

**Purification of rS1, rS1ΔD6 and rS1ΔD56**. The rS1 sequence was taken from *Vibrio vulnificus* (strain CMCP6). rS1, rS1ΔD6 and rS1ΔD56 were expressed in BL21(DE3) cells carrying a N-terminal polyhistidine- and thioredoxin-tag. Cell cultures were grown in LB-medium at 37 °C. At OD$_{600}$ 0.6–0.8, protein expression was induced with 0.5 mM isopropyl β-D-1-thiogalactopyranoside (IPTG) and incubated over night at 20 °C. The culture was harvested and resuspended in 50 mM Tris-HCl (pH 8), 300 mM NaCl, 10 mM imidazole, 10 mM β-mercapto-ethanol. The protein was purified from the soluble fraction with a 5 mL HisTrap column (GE Healthcare) using 50 mM Tris-HCl (pH 8), 300 mM NaCl, 500 mM imidazole, 10 mM β-mercapto-ethanol for elution. Tags were cleaved overnight and at 4 °C using TEV protease and removed via a second HisTrap column. The protein was further purified by size exclusion chromatography on a HiLoad 26/60 S75 column (GE Healthcare) using NMR buffer (25 mM potassium phosphate, 150 mM KCl, 5 mM DTT, pH 7.2).

**Purification of 30S ribosomes and rS1-depleted 30S ribosomes**. 30S ribosomes were purified based on His-tagged proteins L7/L12 following the single step protocol from Ederth et al.[33] Briefly, *E. coli* strain JE28 was grown in LB medium supplemented with 50 μg/mL kanamycin at 37 °C. At OD$_{600}$ 0.6, the culture was harvested by centrifugation at 4000 g for 5 min. The cell pellet was resuspended in lysis buffer (20 mM Tris–HCl pH 7.6, 10 mM MgCl$_2$, 150 mM KCl, 30 mM NH$_4$Cl) supplemented with 0.5 mg/mL lysozyme, 50 μL DNase 1 (2000 U/mL) and cOmplete™ Protease Inhibitor Cocktail tablet (Roche) and further lysed using a Microfluidizer M-110P (Microfluidics, USA) in a final volume of 50 mL. The lysate was clarified by centrifuging twice at 38420 g at 4 °C, 20 min each. For affinity purification, a HisTrap column (Ni$^{2+}$ sepharose pre-packed, 5 mL, GE Healthcare) was connected to an ÄKTA prime chromatography system (GE Healthcare) pre-equilibrated in lysis buffer. After loading the lysate, the column was washed with 10 column volumes of 7 mM imidazole in lysis buffer. The purification of 30SΔS1 ribosomes was performed as described before. To elute the ribosomal S1 protein, the column was washed with 1.5 M NH$_4$Cl in lysis buffer. To elute 30S ribosomes, a decreasing magnesium concentration gradient was applied from 10 mM to 1 mM magnesium. Fractions were concentrated to 1 mL using centrifugal filter units (Vivaspin 20, MWCO 30000) and loaded onto a sucrose gradient of 20% to 50%

sucrose in sucrose gradient buffer (20 mM Tris-HCl, pH 7.6, 300 mM NH$_4$Cl, 5 mM Mg(OAc)$_2$, 0.5 mM EDTA and 7 mM β-mercapto-ethanol). Gradients were centrifuged for 16 h at 100000 g and 4 °C, fractionated with a Piston Gradient Fractionator (BioComp Instruments, Canada) and analyzed by SDS gel electrophoresis (Supplementary Fig. 8). 30S ribosomes were stored in sucrose buffer at 4 °C. Concentration of 30S ribosomes were determined at A$_{260}$ where 1 A$_{260}$ correspond to 72 pmol/mL[34]. To avoid disturbing effects of sucrose, sample concentration was determined in 1:100 dilutions after buffer exchange.

**30S ribosome stability and integrity of NMR samples**. High resolution NMR experiments require that the sample of the macromolecular complex has to be pure from contaminating proteins and nucleic acids, it must be monodispersed and due to the experimental restraints highly concentrated. Purity of the final 30S ribosomes utilized in all subsequent experiments was judged by gel-based analysis in comparison to similarly purified 50S and 70S particles (Supplementary Fig. 8).

Ribosomes were further analyzed by Western Blot using a 6x-His Tag Antibody (Thermo Fischer) that allows selective detection of the modified ribosomal protein L12. The Western Blot with this specific antibody clearly shows that 30S particles are free of impurities of 50S protein components.

Integrity of ribosomes were further analyzed by native composite gel electrophoresis and quantitative mass spectrometry (Supplementary Fig. 9). After acquiring NMR experiments, the NMR samples were analyzed for RNA integrity and 30S ribosome stability. RNA integrity was checked by denaturing polyacrylamide gel electrophoresis (Supplementary Fig. 8). Theoretically, RNA could possibly be degraded by co-purification of RNases during ribosome preparation. Ribosome stability was further analyzed by loading the measured samples on another sucrose gradient (Supplementary Fig. 10).

**Electrophoretic mobility shift assay (EMSA)**. The EMSAs to investigate RNA-S1 binding were performed at constant RNA concentration (4 μM) and increasing amounts of protein (0–10 equivalents) in the absence and presence of adenine. All samples were prepared in EMSA buffer (25 mM potassium phosphate, 75 mM KCl, 5 mM DTT, pH 7.2) and were incubated for 30 min on ice. Bands were separated on continuous gels (8% PAGE) at low powers (<500 mW, 3.5 V/cm) in TAM buffer (40 mM Tris, 20 mM acetic acid, 5 mM MgCl$_2$, pH 8.3). The RNA was visualized upon staining with GelRed$^{TM}$ (Biotium) and subsequent excitation of the bands at 254 nm.

**Native composite gel**. For the analysis of ribosome stability and electrophoretic mobility shift assays, native composite gels were performed which allow the migration of native ribosomes into the gel[35]. Optimized composite gels were composed of 1.76% acrylamide and 0.68% agarose. Therefore, 13.5 mL of 0.92% agarose solution was cooled to 65 °C. Subsequently, 3 mL of prewarmed buffer (0.2 M Tris, 0.125 M acetic acid, 35% (w/v) sucrose) were added to the agarose solution. Then, 1 mL of acrylamide (29:1 acrylamide:bisacrylamide), 8.5 μL of Triton-X100 and MgCl$_2$ (5 mM) was added and heated to 65 °C. Polymerization was induced by adding 100 μL of APS and 20 μL of TEMED and poured immediately into a prewarmed gel chamber. The gel was polymerized for 1 h at 4 °C. A pre-run was conducted at 200 V for 30 min under water cooling. The running buffer (40 mM Tris, 25 mM acetic acid, 5 mM MgCl$_2$) was exchanged.

The 127 nt *add* riboswitch was prepared by the ligation of a 97 nt fragment (G14–G110) with the Cy5-labeled 30 nt RNA (A111-G140) (Dharmacon) as described before[36]. The RNA was pre-incubated with 1.4 equivalents of adenine. Ribosome concentration was increased from 1 to 10 equivalents at a constant RNA concentration of 1.28 μM. All samples were prepared in NMR buffer (25 mM potassium phosphate, 150 mM KCl, 5 mM MgCl$_2$, pH 7.2) and a final glycerol concentration of 10% as loading buffer. Samples were loaded and the gel was run for 1 hour and 20 min at 200 V under water cooling. Bands were visualized by UV shadowing, fluorescence scan and further stained with Coomassie.

**Ribosomal RNA extraction**. rRNA of 30S ribosomes were analyzed by agarose gel electrophoresis. Therefore, rRNA was extracted with 1 vol. eq. PCI. Aqueous phase was mixed with gel loading dye and analyzed by 1% agarose gel electrophoresis. The RNA was visualized upon staining with GelRed$^{TM}$ (Biotium).

**NMR Spectroscopy**. All experiments were performed at 25 °C in NMR buffer (25 mM potassium phosphate, pH 7.2, 150 mM KCl, 5 mM MgCl$_2$) and 5% to 10% D$_2$O. For protein-RNA studies, 5 mM DTT was added. For ligand bound RNA, 1.4 equivalents of $^{13}$C-$^{15}$N-labeled adenine was added. $^{13}$C-$^{15}$N-labeled adenine was synthesized as described[37].

All spectra were referenced to DSS (4,4-dimethyl-4-silapentane-1-sulfonic acid). Nitrogen-15 and Carbon-13 chemical shifts were indirectly referenced using the ratio of the gyromagnetic ratios of proton to $^{15}$N (0.101329118) and $^{13}$C (0.251449530), respectively[38].

For titration experiments of the $^{15}$N-labeled 127 nucleotide *add* adenine-sensing riboswitch with the ribosomal S1 protein, the RNA concentration was set to 40 μM. The unlabeled protein rS1 was stepwise added to the RNA in ratios ranging from 0 to 5.

For the investigation of mRNA-ribosome complexes with the 127 nucleotide ASW, samples were prepared with a concentration of 35 μM ribosomes and RNA, since higher concentrations lead to aggregation of the ribosomes.

NMR experiments were performed on an 800 MHz, 900 MHz or 950 MHz NMR spectrometer equipped with a 5 mm, z-axis gradient $^1$H {$^{13}$C, $^{15}$N} TCI cryogenic probe.

BEST-TROSY (Band-selective Excitation Short-Transient - Transverse relaxation-optimized spectroscopy) experiments were recorded to observe $^1$H, $^{15}$N correlations of $^{15}$N isotope labeled RNA[39]. Spectra were recorded using a modified pulse program[40].

NMR Experiments were analyzed using Bruker Biospin software TopSpin 3.5. Assignments were performed using the software Sparky 3.114[41].

**Microscale thermophoresis**. The 127 nt add riboswitch was prepared by the ligation of a 97 nt fragment (G14–G110) with the Cy5-labeled 30 nt RNA (A111–G140) (Dharmacon) as described before[36]. 30SΔS1 ribosomes were labeled with Monolith NT Protein Labeling kit red-maleimide according to the protocol (NanoTemper Technologies). The Cy5-labeled 127 nt riboswitch (30 nM) was incubated with a 1:1 dilution series of 30S ribosomes (36 μM) and 30SΔS1 ribosomes (31 μM) in the absence and presence of adenine (500 μM). Labeled 30SΔS1 (30 nM) was incubated with a 1:1 dilution series of rS1 (3 μM). The complexes were incubated for 30 min at 25 °C in NMR buffer (25 mM potassium phosphate, pH 7.2, 150 mM KCl, 5 mM MgCl$_2$, 5 mM DTT) supplemented with 0.4 mg/mL BSA (New England Biolabs), 0.05% Tween20 and RNasin (Promega, Germany). The reactions were transferred to hydrophobic capillaries and measured with a microscale thermophoresis (MST) device (NanoTemper Monolith NT.115, Germany). After a primary capillary scan, the thermophoresis experiments were carried out at 25 °C with an excitation and MST power of 40% each. Four scans were recorded for each sample. The analysis was performed using MO.Affinity Analysis and the evaluation strategy T-Jump.

**In vivo assay of adenine-sensing riboswitch regulated gene expression**. BL21 (DE3) E. coli cells were transformed with pUC19 plasmid containing T7 regulatory elements and shifted green fluorescent protein (sGFP) gene under the translational control of the adenine-sensing riboswitch ASW14–122. For a pre-culture, a single colony was picked and added to 5 mL M9 minimal medium containing 100 mg/mL ampicillin and IPTG and incubated in a thermoshaker (37 °C, 120 rpm) overnight. The pre-culture was used to inoculate 50 mL M9/ampicillin main cultures with varying adenine and IPTG concentrations in biological triplicates with a start OD$_{600}$ of 0.1, which was grown in a thermoshaker (37 °C, 120 r.p.m.). The production of sGFP was monitored via fluorescence emission ($\lambda_{ex}$ = 484 nm and $\lambda_{em}$ = 510 nm) with a Tecan Infinite M200 pro microplate reader (Tecan, Männedorf, Switzerland). For error estimation, three biological independent samples were measured in triplicates. Results were normalized to the growth condition of 0 mM adenine and 0 mM IPTG. As additional control, untransformed BL21(DE3) E. coli cells were measured.

**Coupled in vitro transcription-translation assay**. The coupled in vitro transcription-translation assays and the cell extract preparation were performed as previously described[42] to monitor protein expression levels of shifted green fluorescent protein (sGFP). The cell-free protein expression reactions were performed with in-house made S100 cell extracts from E. coli strain BL21 Star$^{TM}$(DE3) and rS1-depleted ribosomes (final concentration 0.46 μM). The S100 extract preparation was adapted from the S30 extract protocol by Schwarz et al.[43] Our preparation differs at the dialysis step. Here, the S100 extract was dialyzed at 4 °C against dialysis buffer (10 mM Tris/acetate (pH 8.2), 14 mM Mg(OAc)$_2$, 60 mM KOAc, 0.5 mM DTT) using a dialysis membrane (MWCO 12-14 kDa, ZelluTrans, Carl Roth, Germany). The dialysis was performed for 3 h and the buffer was exchanged every hour. After dialysis the S100 extract was centrifuged at 40000 rpm and 4 °C for 5 h under vacuum (Beckman Coulter Optima$^{TM}$ L-90 K). The protein is synthesized from a pUC19 plasmid containing T7 regulatory elements. The sGFP gene is under the translational control of the adenine-sensing riboswitch ASW14–122 that contains a weak ribosome-binding site and the start codon AUG. The 30 μL reactions were performed as triplicates in μClear® 384-well flat bottom microplates (Greiner Bio-One, Frickenhausen, Germany) at 25 °C for 2 h. The production of sGFP was monitored via sGFP fluorescence ($\lambda_{ex}$ = 470 nm and $\lambda_{em}$ = 515 nm) with a Tecan Spark® microplate reader (Tecan, Männedorf, Switzerland). For error estimation, one biological sample was measured in triplicates. Native 70S ribosomes were used as positive control.

**Structural modeling of mRNA-ribosome complexes**. The riboswitch structure was modeled into the mRNA decoding channel of the 30S ribosome.

For the structural modeling, the 30S ribosomal pre-initiation complex structure with the PDB ID 5lmn was chosen. The proteins IF1, IF3 and Thx were removed from the structure.

The adenine-sensing riboswitch structure was built as follows: the holo aptamer structure (C13–G83) with the PDB ID 1y26 was used as starting point. Here, the nucleotides C13 and G83 were removed from the structure.

To build the single stranded region of the mRNA, the online tool RNAthread (ROSIE) was used[44,45]. Here, the single stranded mRNA from the 5lmn structure with 20 nucleotides was used as input structure. The nucleotides were replaced in steps of 10–20 nucleotides with the desired ASW sequence. The resulting structure was then renumbered accordingly to the ASW sequence using the renumber_PDB tool.

Sequences were aligned in bins of two, each containing two overlapping nucleotides. The alignment was performed in Pymol. This procedure ensured preservation of the correct backbone geometry. The redundant nucleotides from the alignment were removed manually in text editor. The mRNA in the 5lmn structure was replaced by the modeled ASW structure, maintaining the same position of the GAG triplet (G110–G112) from the Shine–Dalgarno sequence, that base pairs with the rRNA (C1536–C1538).

**Reporting summary**. Further information on research design is available in the Nature Research Reporting Summary linked to this article.

## Data availability

The data needed to evaluate the conclusions in the paper are present in the paper and/or the supplementary information, and are available from the corresponding author upon reasonable request. The structural model that was generated in this study based on the publically available structures 1Y26 and 5MLN is provided in Supplementary Data 1. Source data are provided with this paper.

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

## Acknowledgements
We are grateful to Suparna Sanyal for providing *E. Coli* JE28 cells. We thank Julian Langer for quantitative mass spectrometry analysis. We thank Frank Löhr for help with HNN-COSY experiments. Further, we thank Volker Dötsch for kindly providing the pIVEX-sGFP plasmid. VDJ was supported by Boehringer Ingelheim Fonds. The work was supported by funds from DFG in collaborative research center 902: "Molecular principles of RNA-based regulation" and in Graduate College GRK1986 (CLIC). Work at BMRZ is supported by the state of Hesse (HMWK).

## Author contributions
V.D.J., N.S.Q., S.W. and J.K.B. prepared the samples. V.D.J., N.S.Q. and B.F. performed the NMR experiments. V.D.J. performed analytical gel electrophoresis, EMSAs, composite gels, western blot and sucrose gradients. V.D.J., N.S.Q. and S.W. performed the MST experiments. V.D.J., M.S.D. and M.H. performed binding studies. V.D.J. performed the in vivo assay. J.K.B. performed the coupled transcription-translation assay. B.F. performed rRNA analysis. B.F. and H.S. directed the research. V.D.J., B.F. and H.S. wrote the manuscript. All authors provided critical comments.

## Funding

## Competing interests
The authors declare no competing interests.
