## [Peer Review File · Nature Communications]

Reviewers' Comments:

Reviewer #1:

Remarks to the Author:

The manuscript by de Jesus et al. reports on effector small-molecule-dependent 30S ribosomal subunit binding to a translational adenine-sensing bacterial riboswitch. In the 20 years since their discovery, most biophysical studies on riboswitches have concentrated on their effector binding ("aptamer") domain, ignoring how these gene-regulatory mRNA domains interface with the gene expression machinery. This manuscript takes an important step forward by investigating the role of 30S ribosomal subunits into switching by a riboswitch that functions (ostensibly) by modulating access to the ribosome binding site as a function of intracellular effector concentration. As such, I think the manuscript, which appears experimentally sound, will be of interest not only to experts in riboswitch function, but also more broadly to those interested in gene regulation. Moreover, the NMR experiments are state-of-the-art, and will be of interest to that community, as experimental systems progressively move toward the full biological context.

That said, I think the manuscript as it stands sweeps under the rug a fundamental issue that plagues riboswitch studies, and that is the role of kinetic control in regulation. Thus, I think a more thorough discussion of the implications of the data is warranted prior to publication.

1. Specifically, the starting point of the manuscript is that two states of the fully transcribed riboswitch, apoA and apoB coexist in equilibrium prior to adenine binding, and the small molecule shifts the equilibrium distribution toward one where the 30S can bind. The authors present here and previously, strong evidence that this is true in vitro. However, as far as I can see, there is no evidence presented or cited that the full-length riboswitch exists in pre-transcribed form, in the living cell, in equilibrium with adenine. Given the rather massive literature on transcription-translation coupling in bacteria, and the growing literature on the role of kinetic control in riboswitch function, this premise for the study needs

to be articulated clearly, and if there is any evidence against kinetic control in the function of this RNA, that needs to be discussed at the outset.

2. Indeed, taken at face value, the data presented in this manuscript could indicate a role for kinetic control. Thus, equilibrium adenine binding to the RNA is quoted as 600 nM, while the output (eGFP in this case) seems to achieve 1/2 of its total amplitude change somewhere in the 0.5 mM (1000 times higher concentration). Absent other issues (that may well be discussed), this is evidence for kinetic control, i.e., that the folding and ligand (and 30S) binding of the riboswitch are occurring in comparable timeframes. Thus, the question is one of rates of competing phenomena, in this case in the time window between complete transcription of the minimal aptamer domain competent for adenine binding, and exit from the RNA polymerase of sufficient RNA for the 30S to load. Is there evidence, in the literature or otherwise, that the transcript fully equilibrates in that time window?

3. A further confusing point to be discussed is raised by data in figure 2c. If the K_d for adenine of the riboswitch is 600 nM, but the full 30S binds even the adenine-free riboswitch with $K_d = 320$ nM, then the mRNA would be ON, regardless of adenine occupancy. Moreover, as in point 1 above, this K_d is three or more orders of magnitude smaller than the apparent in vivo sensitivity of the regulator.

Reviewer #2:

Remarks to the Author:

The authors studied the interaction of the adenine-sensitive riboswitch of *Vibrio vulnificus* with the *E. coli* 30S ribosomal subunit using NMR. They show that the ribosomal protein S1 of *Vibrio vulnificus* is necessary to obtain the translational on-state of the riboswitch. The paper is well written and the experiences are well explained and analyzed.

My concern is that they are using the E. coli 30S subunit without really arguing for it. They do not use figure S11. I think the article could benefit from a clear explanation of why they are using the E. coli as a model for this study and why it is warranted. It should also be mentioned in Figure 4 that it is the ribosome of E. coli that is used.

In the discussion, it might be interesting to speculate to what extent the discovery of the use of a ribosomal protein by translational riboswitches is general. Can we predict that it is always S1 that interacts with the riboswitch or that another ribosomal protein could be used? Is this a chaperone activity of S1?

Reviewer #3:

Remarks to the Author:

The manuscript "Switching at the Ribosome: Riboswitches need rProteins as modulators to regulate translation" by Jesus et al. describes the interaction of riboswitches with the translation machinery. Translational riboswitches are cis-acting RNA regulators that modulate the expression of genes during translation initiation. The mechanism is considered as an RNA-only gene-regulatory system inducing a ligand-dependent shift of the population of functional ON- and OFF-states. Based on the investigation of the adenine-sensing riboswitch from *Vibrio vulnificus*, the authors show that ligand binding alone is not sufficient for switching to a translational ON-state but the interaction of the riboswitch with the small ribosomal subunit is indispensable. Only the synergy of binding of adenine and of 30S, in particular protein rS1, induces complete opening of the translation initiation region. The authors think their investigation unravels the intricate dynamic network involving RNA regulator, ligand inducer and ribosome protein modulator during translation initiation.

However, there are still some points in the manuscript, which need to be illustrated or corrected. I only support the publication of it in Nature Communication after the proper revision. All the points are listed below.

1. All the panel labeling in the Figures are lowercase letters, but in the manuscript, all are written as capital letters. For example, Fig. 1A (P2, line 31) should be Fig. 1a. Please correct them.
2. In P3, line 15, "only affected nucleotides up to position -9", please label "-9" in the relevant Figure. In Figure 1 and S1, the nucleotide numbering is 14 to 140, however in Figure S3, the labelling is changed to -15 to +15. It is better to make them consistent in manuscript.
3. In P3-line 13, P4-line 22 and P4-line 25, 30S is written as "30S ribosomal subunit", "30S subunit" and "30S ribosomes" respectively. I'm wondering the difference between them. Please clarify it.
4. In P4-line 20, the authors describe "However, the regulation efficiency diminished to basal levels in the absence of rS1. The switching behavior could be restored upon addition of rS1." However, in Fig 2b, the reviewer didn't see much difference between the absence of rS1 and the addition of S1. Please clarify it.
5. In P4-line 24, one mistake " $K_D = 4000 \pm 800 \mu\text{M}$ " should be " $K_D = 4000 \pm 800 \text{ nM}$ ", Please correct them.
6. As in Figure 1a, stem P5 is located far away from the binding pocket. The switched elements of riboswitch that pair with the sequence of SD is located in stem P4. The experiments in Figure 3b shown that the addition of 30S or 30SΔS1 both generate the complete opening of stem P4, which seems S1 is not necessary for the binding of SD sequence. S1 may contribute a lot to the opening of stem P5, which conform to the function that "S1 is essential for translation initiation, as it recruits the mRNA and facilitates its localization in the decoding center." cited from (NAR, Nucleic Acids Research, 2015, Vol. 43, No. 1 661–673). So only one case with one sequence, it seems not enough to challenge the function of riboswitch as the RNA-only gene-regulatory system.

7. Please correct P6-line 16 "RNA only"

8. Please correct P7-line 43 "cOplete"

Switching at the Ribosome: Riboswitches need rProteins as modulators to regulate translation

Vanessa de Jesus¹, Nusrat S. Qureshi¹, Sven Warhaut¹, Jasleen K. Bains¹, Marina S. Dietz²,

Mike Heilemann², Harald Schwalbe¹, Boris Fürtig^{1*}

REVIEWER COMMENTS	
Reviewer #1	
The manuscript by de Jesus et al. reports on effector small-molecule-dependent 30S ribosomal subunit binding to a translational adenine-sensing bacterial riboswitch. In the 20 years since their discovery, most biophysical studies on riboswitches have concentrated on their effector binding ("aptamer") domain, ignoring how these gene-regulatory mRNA domains interface with the gene expression machinery. This manuscript takes an important step forward by investigating the role of 30S ribosomal subunits into switching by a riboswitch that functions (ostensibly) by modulating access to the ribosome binding site as a function of intracellular effector concentration. As such, I think the manuscript, which appears experimentally sound, will be of interest not only to experts in riboswitch function, but also more broadly to those interested in gene regulation. Moreover, the NMR experiments are state-of-the-art, and will be of interest to that community, as experimental systems progressively move toward the full biological context. That said, I think the manuscript as it stands sweeps under the rug a fundamental issue that plagues riboswitch studies, and that is the role of kinetic control in regulation. Thus, I think a	We thank the reviewer for the positive comments. As follows in the answers to the next comments of the reviewer we have now included a more thorough discussion of the kinetic aspects of regulation by the riboswitch.

more thorough discussion of the implications of the data is warranted prior to publication.	
1. Specifically, the starting point of the manuscript is that two states of the fully transcribed riboswitch, apoA and apoB coexist in equilibrium prior to adenine binding, and the small molecule shifts the equilibrium distribution toward one where the 30S can bind. The authors present here and previously, strong evidence that this is true in vitro. However, as far as I can see, there is no evidence presented or cited that the full-length riboswitch exists in pre-transcribed form, in the living cell, in equilibrium with adenine. Given the rather massive literature on transcription-translation coupling in bacteria, and the growing literature on the role of kinetic control in riboswitch function, this premise for the study needs to be articulated clearly, and if there is any evidence against kinetic control in the function of this RNA, that needs to be discussed at the outset.	We thank the reviewer for pointing out this very important concept in riboswitch control. To clarify the question raised, we want to refer to the paper of Lemay et. al. (2011) who have studied two adenine-sensing riboswitches that show fundamentally different modes of operating, one from V. vulnificus and B. subtilis. The pbuE riboswitch from B. subtilis is a transcriptional riboswitch, while the add riboswitch V. vulnificus controls translation initiation. The regulation of both riboswitches relies on fundamentally different mechanisms. The pbuE riboswitch operates under kinetic control and has to fold and to bind adenine co-transcriptionally. In addition, transcriptional pausing is crucial. In contrast, it has been shown that translation activation by the add riboswitch is under thermodynamic control. The add riboswitch can bind adenine after the transcription of the mRNA. In vitro translation assays using DNA and mRNA showed that the gene regulation is not coupled in the add riboswitch. The addition of adenine to the transcribed mRNA results in a 3-fold higher expression level. In addition, in vivo assays using transcriptional and translational constructs were performed by Lemay et al. Here, only the translational fusion showed 3-fold higher expression levels. These in vivo assays clearly demonstrated riboswitch regulation in vivo. As the add riboswitch can bind adenine after the mRNA was transcribed, the author proposed that the add riboswitch reversibly binds adenine at equilibrium. Moreover, both ON and OFF structures harbor similar free energies, supporting the rapid fluctuation between both states. Taken together, gene regulation of the add riboswitch is not dependent on transcription-translation coupling. We thank the reviewer for the comment and apologize that this was not clarified. Given this discussion, we added the following discussion to the manuscript: “The add riboswitch can bind adenine post-transcriptionally and operates

	under thermodynamic control, as shown by in vitro and in vivo studies¹⁴. Gene regulation of the add riboswitch is independent from the coupling of transcription and translation. It is clear that the kinetics of ligand binding and structural switching between apoA and apoB has to occur on a timescale faster than translation initiation^{12,15}.
2. Indeed, taken at face value, the data presented in this manuscript could indicate a role for kinetic control. Thus, equilibrium adenine binding to the RNA is quoted as 600 nM, while the output (eGFP in this case) seems to achieve 1/2 of its total amplitude change somewhere in the 0.5 mM (1000 times higher concentration). Absent other issues (that may well be discussed), this is evidence for kinetic control, i.e., that the folding and ligand (and 30S) binding of the riboswitch are occurring in comparable timeframes. Thus, the question is one of rates of competing phenomena, in this case in the time window between complete transcription of the minimal aptamer domain competent for adenine binding, and exit from the RNA polymerase of sufficient RNA for the 30S to load. Is there evidence, in the literature or otherwise, that the transcript fully equilibrates in that time window?	We thank the reviewer for these comments. The in vivo assay was performed using the sGFP gene under the translational control of the adenine-sensing riboswitch in minimal medium. The experiments were conducted in E. coli and not in V. vulnificus for practical reasons. Here, adenine was added to the cultures. The free adenine concentration in the in vivo experiments is not known as adenine is used in various metabolism pathways in the cell. As mention above, we agree that folding and ligand binding has to occur at comparable time frames. However, we refrain from discussing our data in light of a kinetic control mechanism, as we cannot determine the active free adenine concentration within the bacterial cells in our in vivo assays. As described above, the riboswitch operates under thermodynamic control. Therefore, the adenine-dependent equilibration between the ON- and OFF-states must occur faster than translation initiation. For this adenine-dependent riboswitch it was shown by Reining et. al. that the rate limiting step in regulation is dependent on the RNA refolding rate, which is independent from adenine concentration and that this occurs on the timescale of the initiation of translation. We thank the reviewer for these comments, and we added the following description in the manuscript: “It was shown that the rate limiting step in regulation is dependent on the RNA refolding rate, which is independent from adenine concentration¹². “
3. A further confusing point to be discussed is raised by data in	We thank the reviewer for this comment and acknowledge that the

figure 2c. If the K_d for adenine of the riboswitch is 600 nM, but the full 30S binds even the adenine-free riboswitch with $K_d = 320$ nM, then the mRNA would be ON, regardless of adenine occupancy. Moreover, as in point 1 above, this K_d is three or more orders of magnitude smaller than the apparent in vivo sensitivity of the regulator.	description was too brief. Explained by the studies presented in this manuscript we want to highlight that there is a difference between an adenine-free riboswitch bound to the 30S ribosome and a functional ON-state that is represented by a riboswitch bound to the ribosome. Both “ligands” (adenine and 30S ribosome) have to bind synergistically in order to induce a functional ON-state. Only then, the SD sequence is completely released and hybridizes with the aSD sequence of the 16S rRNA leading to correct positioning of the start codon. We thank the reviewer for this very crucial comment and added the following sentence in the manuscript “The observation that adenine binds more weakly to the ASW than the ribosome hints at a structural and functional synergistic effect of the inducer ligand and the ribosome.”
Reviewer #2	
The authors studied the interaction of the adenine-sensitive riboswitch of Vibrio vulnificus with the E. coli 30S ribosomal subunit using NMR. They show that the ribosomal protein S1 of Vibrio vulnificus is necessary to obtain the translational on-state of the riboswitch. The paper is well written and the experiences are well explained and analyzed. My concern is that they are using the E. coli 30S subunit without really arguing for it. They do not use figure S11. I think the article could benefit from a clear explanation of why they are using the E. coli as a model for this study and why it is warranted.	We thank the reviewer for the positive feedback. We agree that this point needs explanation. For practical and biosafety issues, we are unable to access the ribosome of V. vulnificus, but with the E. coli system, we have a suitable supplement. As presented in the now referenced figure S11, the 3'-end of the 16S ribosomal RNA that harbors the anti-Shine-Dalgarno sequence is nearly identical (98%) between E. coli and V. vulnificus. Further, the domain organization of ribosomal protein S1 is identical in both species, and the homology is also very high (approx. 86%). At the same time, both bacteria control gene regulation preferably on the level translation. This was a mistake from our side and we added the following paragraph in the manuscript: As the cognate ribosome from V. vulnificus is not accessible for practical and biosafety reasons, the interaction is studied here with the homologous E. coli ribosome. The 3'-end of the 16S rRNA in E. coli which harbors the aSD sequence is nearly identical (identity = 98%) to that of V. vulnificus (Figure

	S11) supporting the validity of using E. coli ribosomes to study the molecular basis of translation initiation for the add riboswitch from V. vulnificus.
It should also be mentioned in Figure 4 that it is the ribosome of E. coli that is used.	We acknowledge the comment and added the description in the figure caption.
In the discussion, it might be interesting to speculate to what extent the discovery of the use of a ribosomal protein by translational riboswitches is general. Can we predict that it is always S1 that interacts with the riboswitch or that another ribosomal protein could be used? Is this a chaperone activity of S1?	We agree with the reviewer that a deeper discussion is interesting, giving the fact, that the S1 is predestinated from its placement at the 30S ribosome to interact with 5'UTRs, from its chaperoning activity and from its functional role within mRNA delivery to generally interact with any translational riboswitch. Further it could be shown that rS1 interacts with riboswitches also in isolation in a way that induces partially single stranded conformations. Therefore we have included the following statement within the manuscript: “Here, rS1 exerts its RNA chaperoning function²⁰ directly at the ribosome, fulfilling its role in mRNA delivery by keeping the ribosome binding site on the mRNA in a single stranded conformation. As it is shown that rS1 promotes partial mRNA unfolding also in other riboswitches²⁸, facilitating single-stranded conformations within the effector domains of translational riboswitches might be a general function of rS1. However, for the ASW, the observed unfolding of this part of the riboswitch is prerequisite for productive incorporation of the mRNA into the 30S ribosome.” Given this, we think that S1 is the natural candidate for this role. However, we cannot exclude that in other cases, other ribosomal proteins or ribosome associated proteins fulfill a similar functional role. However, we do not want to speculate at this point.
Reviewer #3	
The manuscript “Switching at the Ribosome: Riboswitches need rProteins as modulators to regulate translation” by Jesus et al. describes the interaction of riboswitches with the translation machinery. Translational riboswitches are cis-acting RNA regulators that modulate the expression of genes during	We thank the reviewer for the positive feedback. The panel labeling was a mistake from our side, which was corrected. We thank the reviewer for the observation.

translation initiation. The mechanism is considered as an RNA-only gene-regulatory system inducing a ligand-dependent shift of the population of functional ON- and OFF-states. Based on the investigation of the adenine-sensing riboswitch from Vibrio vulnificus, the authors show that ligand binding alone is not sufficient for switching to a translational ON-state but the interaction of the riboswitch with the small ribosomal subunit is indispensable. Only the synergy of binding of adenine and of 30S, in particular protein rS1, induces complete opening of the translation initiation region. The authors think their investigation unravels the intricate dynamic network involving RNA regulator, ligand inducer and ribosome protein modulator during translation initiation. However, there are still some points in the manuscript, which need to be illustrated or corrected. I only support the publication of it in Nature Communication after the proper revision. All the points are listed below. 1. All the panel labeling in the Figures are lowercase letters, but in the manuscript, all are written as capital letters. For example, Fig. 1A (P2, line 31) should be Fig. 1a. Please correct them.	
2. In P3, line 15, “only affected nucleotides up to position -9”, please label “-9” in the relevant Figure. In Figure 1 and S1, the nucleotide numbering is 14 to 140, however in Figure S3, the labelling is changed to -15 to +15. It is better to make them consistent in manuscript.	We thank the reviewer for this very crucial comment. In order to prevent this misunderstanding for potential readers, we have modified the Figure S3 and changed the labeling from “-15 to +15” to the nucleotide numbering “14 to 140”. In addition, we also added the missing numbering “105” and “110” in Fig. 1. We apologize for the confusion and modified the paragraph: “The footprint of bound 30S ribosomal subunit comprises 30 nucleotides that start 15 nt upstream of the start codon (position -15)¹⁴. The adenine-induced opening of RNA structure in ASW¹⁴⁻¹⁴⁰, however, only affected nucleotides up to position -9, nucleotide A111 (Fig. S1).”
3. In P3-line 13, P4-line 22 and P4-line 25, 30S is written as “30S	We apologize for the confusion. We thank the reviewer for this comment

ribosomal subunit”, “30S subunit” and “30S ribosomes” respectively. I’m wondering the difference between them. Please clarify it.	and we have rephrased “30S ribosomal subunit”, “30S subunit” and “30S ribosomal particles” to “30S ribosome” throughout the manuscript. The changes are highlighted in the manuscript.
4. In P4-line 20, the authors describe “However, the regulation efficiency diminished to basal levels in the absence of rS1. The switching behavior could be restored upon addition of rS1.” However, in Fig 2b, the reviewer didn’t see much difference between the absence of rS1 and the addition of S1. Please clarify it.	We thank the reviewer for this comment. To enhance the visibility of the changes we modified Figure 2 and included the factor in Figure 2b. Further, we added the fold changes into the text, it now reads: “However, the regulation efficiency diminished to basal levels in the absence of rS1, decreasing by a factor of 1.6 upon removal of rS1.”
5. In P4-line 24, one mistake “KD = 4000 ± 800 μM” should be “KD = 4000 ± 800 nM”, Please correct them.	We are grateful for this very important remark. We corrected the manuscript: “The binding constants of apo and holo ASW¹⁴⁻¹⁴⁰ to 30SΔS1 ribosomes were $K_D = 4000 \pm 800$ nM and 2200 ± 200 nM, respectively.”
6. As in Figure 1a, stem P5 is located far away from the binding pocket. The switched elements of riboswitch that pair with the sequence of SD is located in stem P4. The experiments in Figure 3b shown that the addition of 30S or 30SΔS1 both generate the complete opening of stem P4, which seems S1 is not necessary for the binding of SD sequence. S1 may contribute a lot to the opening of stem P5, which conform to the function that “S1 is essential for translation initiation, as it recruits the mRNA and facilitates its localization in the decoding center.” cited from (NAR, Nucleic Acids Research, 2015, Vol. 43, No. 1 661–673).	We thank the reviewer for these comments. The addition of 30S or 30SΔS1 to the ASW holo lead to the opening of helix P4. It is correct that the Shine-Dalgarno sequence, located in P4, is now accessible. However, this is not sufficient for binding of the 30S ribosome, as steric clashes with P5 would prohibit pairing to the 16S rRNA (aSD). It is essential to open P5 to allow the interaction of SD and aSD. The model (Fig. 4) showed that only if helix P5 was single stranded steric clash between rproteins and mRNA could be avoided at the entrance of the tunnel. We agree that it is not unexpected that S1’s chaperone function facilitates riboswitch binding. However, we want to point out that our work shows for the first time riboswitch structure and function in complex with the translation machinery experimentally. Here, to achieve complete opening of the SD sequence, the interaction with the translation machinery and in particular with the rS1 protein are essential. A translational ON-state in the complex between the riboswitch and the 30S ribosome was only achieved when both, the inducer ligand and the modulator rS1 were present.

So only one case with one sequence, it seems not enough to challenge the function of riboswitch as the RNA-only gene-regulatory system.	We thank the reviewer for the comment. In this case, we respectfully disagree with the reviewer. We thank the reviewer for the comment. In this case, we respectfully disagree with the reviewer. This manuscript extends the view of RNA-only hypothesis to explain the function of translational riboswitches. In a similar way, for transcriptional riboswitches protein factors including NusA and NusG have been reported to influence the riboswitch function. These previous publications relied also only on “one case with one sequence” (e.g. Wickisier Moll Cell 2005). We here provide the basis to understand the requirements for additional, ribosome-associated factors in translational riboswitches and even show a direct interaction between the riboswitch and rS1. We are sure that this manuscript will initiate for future research on additional riboswitches and their interplay with the translational machinery.
7. Please correct P6-line 16 “RNA only”	We thank the reviewer for the observation. This was a mistake from our side, which was corrected: “Further, it challenges the “RNA-only” model of riboswitch function, as translational regulation only occurs upon concerted yet dynamic interaction of mRNA, 30S ribosomes and associated modulator proteins.”
8. Please correct P7-line 43 “cOplete”	We acknowledge the comment. We changed the sentence to: “The cell pellet was resuspended in lysis buffer (20 mM Tris–HCl pH 7.6, 10 mM MgCl₂, 150 mM KCl, 30 mM NH₄Cl) supplemented with 0.5 mg/mL lysozyme, 50 μL DNase 1 (2000 U/mL) and cOplete™ Protease Inhibitor Cocktail tablet (Roche) and further lysed using a Microfluidizer M-110P (Microfluidics, USA) in a final volume of 50 mL.”

Reviewers' Comments:

Reviewer #1:

Remarks to the Author:

The revised MS satisfies my comments, and is appropriate for publication as is.